# The Biology of Colicin M and Its Orthologs

**DOI:** 10.3390/antibiotics10091109

**Published:** 2021-09-14

**Authors:** Dimitri Chérier, Delphine Patin, Didier Blanot, Thierry Touzé, Hélène Barreteau

**Affiliations:** 1Institute for Integrative Biology of the Cell (I2BC), CEA, CNRS, Université Paris-Saclay, 91198 Gif-sur-Yvette, France; dimitri.cherier@gmail.com (D.C.); delphine.patin@i2bc.paris-saclay.fr (D.P.); didier.blanot@gmail.com (D.B.); thierry.touze@universite-paris-saclay.fr (T.T.); 2Sanofi-Aventis France, External Manufacturing CHC, 82 Avenue Raspail, 94250 Gentilly, France

**Keywords:** colicins, colicin M, peptidoglycan, lipid II, antibacterials

## Abstract

The misuse of antibiotics during the last decades led to the emergence of multidrug resistant pathogenic bacteria. This phenomenon constitutes a major public health issue. Consequently, the discovery of new antibacterials in the short term is crucial. Colicins, due to their antibacterial properties, thus constitute good candidates. These toxin proteins, produced by *E. coli* to kill enteric relative competitors, exhibit cytotoxicity through ionophoric activity or essential macromolecule degradation. Among the 25 colicin types known to date, colicin M (ColM) is the only one colicin interfering with peptidoglycan biosynthesis. Accordingly, ColM develops its lethal activity in *E. coli* periplasm by hydrolyzing the last peptidoglycan precursor, lipid II, into two dead-end products, thereby leading to cell lysis. Since the discovery of its unusual mode of action, several ColM orthologs have also been identified based on sequence alignments; all of the characterized ColM-like proteins display the same enzymatic activity of lipid II degradation and narrow antibacterial spectra. This publication aims at being an exhaustive review of the current knowledge on this new family of antibacterial enzymes as well as on their potential use as food preservatives or therapeutic agents.

## 1. Colicins among Bacteriocins

Bacteriocins are ribosomally synthesized antimicrobial proteins or peptides produced by both Gram-positive and Gram-negative bacteria, which are auto-immune to the effects of their own bacteriocin [1,2]. They consist of a large and heterologous group of toxins, displaying generally narrow antimicrobial spectra. Not required for growth, they are used to outcompete other microorganisms for the limiting nutrients in the environment [3,4]. Although they have antibacterial properties, bacteriocins are not considered as antibiotics. They are synthesized during the primary growth phase while the antibiotics produced by bacteria are secondary metabolites [1,5].

Among Gram-negative bacteria, the *Enterobacteriaceae* are the major producers of bacteriocins, which are classified in two categories: low molecular weight bacteriocins or microcins (less than 10 kDa), and high molecular weight bacteriocins including colicins produced by *Escherichia coli* (25 to 80 kDa). Both microcins and colicins have a narrow spectrum of activity, being active against bacterial strains phylogenetically related to the producer. Although they are similar in many ways, microcins and colicins can be distinguished from each other by several key features besides molecular mass difference:

- Microcin secretion, unlike colicin secretion, is not lethal to the producing cell [6].

- Microcins are encoded by typical gene clusters carried by plasmids or by the chromosome; these clusters include genes required for microcin biosynthesis (in some cases encoding a precursor peptide and post-translational modification enzymes), microcin secretion and self-immunity [7]. By contrast, almost all colicins are plasmid-encoded, and genes are mostly located in operons. Actually, a typical colicin gene cluster contains additional genes beside the toxin-encoding gene, one gene encoding the cognate immunity protein and, in many cases, another gene involved in the secretion of the colicin. The product of the immunity gene, located immediately downstream of the toxin-encoding gene, protects the toxin-producing cells both from auto-inhibition and from the same toxin produced by neighboring cells [8]. The usually small immunity protein may be released as tightly bound with the C-terminal domain of the toxin, as it is the case for nuclease colicins [9]. Finally, the third gene from the cluster encodes a lysis protein involved in the release of the colicin in the environment, by modification of the cell envelope structure, leading often to cell death [9,10]. So, whereas most microcins are shown or proposed to bear post-translational modifications, colicins are not post-translationally modified proteins.

- Colicin production, but not microcin production, is mainly induced via the DNA repair network, called the SOS regulatory system [11].

- Their cellular targets are different. Colicins are endowed with either nuclease activity, pore-forming activity or inhibitory activity of peptidoglycan synthesis. However, nuclease colicins hydrolyze DNA or RNA strands whereas microcins target the enzymes responsible for DNA/RNA structure or synthesis. Moreover, the inhibition of peptidoglycan synthesis has never been described for microcins.

Colicins, like microcins, have a three-step mode of action: they first parasitize a receptor on the outer membrane of the target bacteria, then they are translocated inside the target cell where they can eventually exert their lethal action. Translocation through the outer membrane is carried out using multiprotein complexes involved in macromolecular transport through the bacterial envelope, either Tol or Ton system. A possible classification of colicins is based on the translocation system they use. Thus, colicins from groups A and B are imported by using the Tol and Ton systems, respectively. Group A is made up of colicins A, E1 to E9, K, L, N, S4, U and Y; group B of colicins B, D, H, Ia, Ib, M, 5 and 10. Colicins can also be classified according to their mode of action: as mentioned above, ionophoric colicins exert their lethal action through the formation of pores in the inner membrane of the target cell, while enzymatic colicins act through the hydrolysis of essential macromolecules (DNA, ribosomal RNA, transfer RNA). Colicin M is a particular case since it interferes with peptidoglycan metabolism. The classification, nomenclature and biology of colicins have been excellently reviewed by others [9,12].

Colicins share a characteristic modular organization composed of three domains, each devoted to a specific step of their lethal action. The central receptor-binding domain is needed for binding a protein at the surface of the target cell, the N-terminal domain is involved in the translocation through the outer membrane, and the C-terminal domain carries the pore-forming or macromolecule degradation activity [9]. The three-dimensional structures of several colicins support their modular organization, generally displaying elongated shapes where each domain is clearly isolated from the others by α-helices. Only colicins M, B, N et S4 show more compact structures (Figure 1).

## 2. Colicin M

### 2.1. General Characteristics

Colicin M (ColM) contains 271 amino acid residues; its molecular mass is 29 kDa, which makes it one of the smallest colicin identified to date [13,14]. The ColM-encoding gene (*cma*) is carried by the pColBM plasmid, which also encodes colicin B (ColB). The *cma* and *cmi* genes, encoding ColM and its immunity protein (Cmi), respectively, are located downstream of the *cba* and *cbi* genes, which encode ColB and its cognate immunity protein, respectively [15,16]. The transcription of *cma* is under the control of the SOS promoter located upstream of *cba*. The *cbi* and *cmi* genes are located on the opposite strand with respect to the colicin-encoded genes and are under the control of their own promoter. No lysis protein has been identified for ColM. Its release into the outside medium is performed by a yet unknown mechanism.

For the reception and translocation steps, ColM parasitizes the FhuA receptor [17,18,19] and the Ton system [17,20,21]. After its passage through the outer membrane, ColM exerts its lethal action through the inhibition of peptidoglycan biosynthesis, which eventually leads to bacterial lysis [22,23]. The reception, translocation and enzymatic activity steps of ColM are supported by its three-domain organization [24].

### 2.2. Tridimensional Structure

The 3D structure of ColM has been determined in 2008 [24] and has allowed to delineate the three domains involved in the reception, translocation and activity steps (Figure 2). As a matter of fact, ColM presents a compact structure as compared to other colicins whose structures have been solved (Figure 2): the three domains are separated from one another by a few amino acid residues only. The N-terminal domain is composed of residues 1–35. As in most colicins, this domain is not structured. It contains the TonB box (E^2^TLTV^6^), which specifically interacts with the translocation system TonB/ExbB/ExbD [20]. The central domain is composed of residues 36–140, which form a globular domain consisting of six α-helices (α1–α6). Finally, the C-terminal domain is composed of residues 141–271. It presents an elongated structure composed of both α-helices and a β-pleated sheet (β3, β4, β5 and β8 stands). The catalytic domain is organized in a half, open β-barrel, which presents structural homology with the β-domain of the Hia autotransporter of *Haemophilus influenzae* [25]. However, the delineation of the catalytic domain according to the 3D structure is not consistent with the minimal length of the catalytic domain as determined through truncation experiments, being composed of residues 122–271 [26].

### 2.3. Mode of Action

#### 2.3.1. Binding to FhuA

In order to bind its target cell, ColM parasitizes the FhuA protein from the outer membrane. FhuA is involved in the ferrichrome uptake [27]. It is also the target of bacteriophages T1, T5 and φ80 [18]. ColM binding involves its central domain, and particularly the α1 helix [28]. The deletion of this helix (ColM Δ38–46) indeed prevents ColM binding to FhuA, while the catalytic activity is not affected since the variant is still active against *E. coli* upon bypassing the import machinery by osmotic shock. The L3, L7 and L8 loops from FhuA, which are present at the outer surface, are essential for ColM activity [29].

Contrary to ColIa, which recruits two copies of the Cir receptor to ensure both reception and translocation steps [30], ColM seems to utilize only one copy of FhuA to perform these steps [31]. Indeed, when ColIa was administered to sensitive cells at a high concentration (molecular ratio ColIa:Cir, 170:1), a strong decrease of cytotoxicity was observed, which was explained by the Cir receptors saturation with administrated ColIa molecules. Consequently, no Cir receptors remained available to perform the translocation. Similar experiments carried out with ColM did not lead to a decrease of cytotoxicity, thereby suggesting that ColM utilizes the same copy of FhuA for the sequential reception and translocation events. This difference can be related to the length of the translocation domains of ColIa and ColM, 225 and 35 residues, respectively. Indeed, once bound to a first Cir receptor, ColIa can recruit a second Cir protein thanks to its long translocation domain. Conversely, the small translocation domain of ColM does not allow the recruitment of a second copy of FhuA posterior to its reception [31].

#### 2.3.2. Translocation

Once bound to the FhuA receptor, the translocation of ColM is mediated by its N-terminal domain [32]. Indeed, the proteolytic digestion of ColM by proteinase K led to the formation of a protein devoid of its 24 first residues. This truncated variant was still capable of binding to FhuA, as demonstrated by the inhibition of the cytotoxic action of phage T5. Moreover, its cytotoxic activity was also preserved as long as it was addressed to the periplasm through osmotic shock. This N-terminal part contains a sequence, the TonB box, enabling ColM to recruit the TonB/ExbB/ExbD import system [14]. FhuA also contains a TonB box in its N-terminal plug domain, which is essential for ColM translocation since mutations in this sequence rendered the cells resistant to ColM [33]. However, suppressive Q160L, Q160K and R158L mutations in TonB restored the functionality of a TonB box FhuA mutant (I9P) [33]. These results suggested a direct interaction between FhuA and TonB, which was further confirmed by the resolution of a 3D structure of FhuA in complex with the C-terminal domain of TonB (Figure 3) [34].

The involvement of the TonB box of ColM in the translocation has been shown by a similar approach [20]. Variants of ColM with mutation in TonB box: L4P, L4N, V6E, V6G and V6R, did not display cytotoxic activity. Nevertheless, ColM sensitivity was restored upon expression of the Q160L TonB protein. The fact that this suppressive mutation restored sensitivity highlighted a direct interaction between ColM and TonB [20]. As a conclusion, the TonB-FhuA and TonB-ColM interactions are essential for the uptake of ColM.

This set of data have allowed Zeth and co-workers to propose a model for the translocation of ColM [24]. According to this model, ColM first binds to FhuA, which then interacts with TonB via its TonB box. This interaction would induce a conformational change of the receptor leading to pore opening, either due to a change of the plug domain itself or through the plug departure from the barrel. The N-terminal domain of ColM can then cross the receptor and interact with the TonB protein in its turn. The energy brought by the Ton system would allow ColM to pass across the FhuA barrel. However, this passage probably requires the unfolding of ColM. Then, to exert its lethal action, ColM must undergo a refolding/maturation process once in the periplasm.

#### 2.3.3. Maturation

Mutant strains that are fully resistant to ColM display mutations in *fhuA*, *tonB*, *exbB* or *exbD* genes. These mutants become ColM sensitive again when the bacteriocin was addressed to the periplasm by osmotic shock in order to bypass the reception/translocation system. However, there exist other mutants, called *tolM*, which were found to resist to ColM even though the colicin was addressed to the periplasm by osmotic shock, suggesting the involvement of one or several other proteins different from the reception/translocation ones, which were essential for ColM activity [35]. The latter mutations occurred in the *fkpA* gene [36]. FkpA is a periplasmic protein involved in the folding of periplasmic and outer membrane proteins [37]. It consists of an N-terminal domain (residues 15 to 114) displaying a chaperone activity and a C-terminal domain (115–224) bearing peptidylprolyl *cis-trans* isomerase (PPIase) activity. Both domains function independently [38] and are essential for ColM activity [39]. FkpA seems to exist only as a dimer [40], with dimerization taking place at the level of the chaperone domain (Figure 4).

It was hypothesized that, during translocation, ColM was unfolded; FkpA would thus be involved in the refolding of ColM. The involvement of FkpA has been demonstrated in vitro since the incubation of urea-denatured ColM with purified FkpA restored ColM cytotoxic activity. This effect was no longer observed in the presence of FK506, which inhibits the PPIase activity of FkpA [36]. Therefore, the PPIase activity appears to be essential for the refolding/maturation of ColM. In order to identify the peptide bond onto which FkpA exerts its isomerase activity, all proline residues of ColM have been individually mutated into alanine [41]. This study allowed the identification of the Phe^175^-Pro^176^ bond as the potential target of FkpA. Indeed, the P176A mutation provoked a strong decrease of ColM cytotoxic activity. Nevertheless, this mutant displayed a residual cytotoxic activity, which was FkpA-independent as long as the protein was addressed to the periplasm by osmotic shock, whereas the wild-type ColM was totally FkpA-dependent in the same conditions. In contrast, the residual activity of the P176A ColM variant remained FkpA-dependent when it penetrated the cell via FhuA or when it was addressed to the periplasm via the Sec system (by fusing the OmpA signal sequence to the N-terminus of ColM), both modes of transport implying ColM unfolding [41]. The latter observation suggested that the isomerisation of the Phe^175^-Pro^176^ bond was not the only action of FkpA towards ColM [26,36].

Whatever the manner by which the wild-type ColM reaches the periplasm (FhuA/Ton, osmotic shock or Sec system), its activity is FkpA-dependent [26,36,39]. However, two studies have shown that a truncated form of ColM, corresponding to its isolated catalytic domain, exerted a cytotoxic activity in an FkpA-independent fashion as long as the FhuA/Ton system was bypassed [26,39]. Interestingly, this isolated catalytic domain displayed a 50-fold higher enzymatic activity of lipid II hydrolysis as compared to the full-length protein [26]. This suggested that the catalytic domain possessed a spatial conformation that was different to the corresponding domain within the full-length protein. The maturation process by FkpA may thus be responsible for this conformational change.

#### 2.3.4. Activity

ColM cytotoxic activity was first observed by Braun and co-workers in 1974 [13]. In their study, the treatment of *E. coli* cells with ColM resulted in a rapid lysis, which was correlated with the release of periplasmic proteins to the outside medium. The observation of treated cells by electronic microscopy revealed the formation of bulges at their surface before lysis. The same treatment in the presence of 15% sucrose resulted in the formation of spheroplasts. These observations suggested that ColM was targeting peptidoglycan since a similar behaviour was observed with penicillin or lysozyme treatments. 

Peptidoglycan is an essential and specific component of the bacterial cell wall, surrounding the cytoplasmic membrane of almost all bacteria. Its main function is to provide cells with protection against internal osmotic pressure. Peptidoglycan is composed of linear glycan chains (made up of alternating *N*-acetylglucosamine (GlcNAc) and *N*-acetylmuramic acid (MurNAc) residues linked in β-1,4) cross-linked by short peptides. Peptidoglycan biosynthesis (Figure 5) is a complex process that involves ca. 20 reactions taking place in the cytoplasm (synthesis of nucleotide precursors), the plasma membrane (synthesis of lipid-linked precursors, lipid I and lipid II, and translocation of lipid II to the outer side of the membrane) and the outer side of the cell (polymerization and maturation reactions). The membrane steps leading to the formation of lipid-linked precursors involve a key lipid, undecaprenyl phosphate (C_55_-P) (Figure 6), needed for the passage of the hydrophilic GlcNAc-MurNAc-peptide monomeric moiety across the inner membrane. Two lipid-linked intermediates are thus produced in the course of peptidoglycan biosynthesis: lipid I (C_55_-PP-MurNAc-pentapeptide) and lipid II (C_55_-PP-MurNAc(-pentapeptide)-GlcNAc), the latter being the last peptidoglycan precursor (Figure 6). After lipid II translocation towards the periplasmic side of the inner membrane, the GlcNAc-MurNAc-peptide moiety is incorporated into the nascent peptidoglycan macromolecule and the lipid carrier is recycled into a pyrophosphorylated form (C_55_-PP). (For comprehensive reviews about peptidoglycan biosynthesis, see [42,43,44,45]).

ColM has been later identified as inhibiting peptidoglycan biosynthesis [22] since, upon treatment with ColM, the cells stopped incorporating radiolabeled diaminopimelate (A_2_pm), which is a specific amino acid to the peptidoglycan peptide stem. Moreover, ColM also causes an arrest of synthesis of the O-antigen moiety from the lipopolysaccharides [46]. Because the C_55_-P lipid carrier is commonly used for O-antigen and peptidoglycan biosynthesis, it was suggested that C_55_-P recycling might be targeted by ColM [46,47]. The exact target of ColM was identified by El Ghachi and co-workers in 2006 [23]. In this study, the effect of ColM was tested in vitro on each enzymatic step of peptidoglycan biosynthesis implying the C_55_-P carrier lipid. ColM was then demonstrated to hydrolyze both lipids I and II into two products identified as undecaprenol (C_55_-OH), and 1-PP-MurNAc-pentapeptide for lipid I or 1-PP-MurNAc(-pentapeptide)-GlcNAc for lipid II, explaining the arrest of peptidoglycan biosynthesis and C_55_-P recycling (Figure 5). The hydrolysis activity of ColM was shown to be dependent on magnesium ions. A kinetics of incorporation of radiolabeled A_2_pm into ColM-treated cells showed a rapid arrest of peptidoglycan synthesis, which was correlated with an accumulation of UDP-MurNAc-pentapeptide and a decrease of lipid I and lipid II amounts. At the same time, the accumulation of C_55_-OH, 1-PP-MurNAc-pentapeptide and 1-PP-MurNAc-pentapeptide-GlcNAc was observed [23]. 

Lipids I and II are sequentially synthesized on the cytoplasmic face of the inner membrane, but only lipid II is accessible on the periplasmic face after its translocation across the membrane. The cellular site of action of ColM was therefore not obvious, since ColM was able to hydrolyze both lipid I and lipid II. However, numerous pieces of evidence allow to conclude that the target of ColM is lipid II on the periplasmic face of the inner membrane: (i) ColM must come from the outside medium to be active [48], while its overproduction in the cytoplasm had no lethal effect. (ii) The immunity protein, Cmi, possesses a transmembrane anchor and a periplasmic domain [49]. (iii) The periplasmic expression of ColM is lethal for the producing cell, except if the cell co-expresses Cmi [39]. The accumulation of 1-PP-MurNAc-pentapeptide observed in treated cells would result from the degradation of 1-PP-MurNAc(-pentapeptide)-GlcNAc by the NagZ glucosaminidase, which is an enzyme involved in the recycling of peptidoglycan fragments [23]. Therefore, the cellular target of ColM was acknowledged to be lipid II. 

Site-directed mutagenesis of ColM has allowed the identification of essential residues involved in the hydrolysis of lipid II [26]. The mutation of residues D226, Y228, D229, H235 or R236 into alanine leads to a dramatic decrease of the in vitro enzymatic activity (<2% of the wild-type activity) and these variants do not display any cytotoxic activity (Table 1).

Among the few ColM orthologs that have been identified to date (see below) in *Pseudomonas*, *Pectobacterium, Klebsiella* and *Bulkholderia* species [50,51,52,53], the sequence D^226^KYDFNASTHR^236^ is globally conserved. According to the structure, these residues are exposed at the surface of ColM (Figure 7). D226 seems to play a crucial role in the activity of ColM, since the mutations D226A, D226E and D226N lead to an almost total loss of activity [26,28]. These results suggested that the carboxyl group of the lateral chain of D226 necessitates a very accurate positioning so that lipid II hydrolysis may take place. This does not seem to be the case for D225 and D229, which are also conserved. Although the essential residues for catalysis have been identified, the exact mechanism is still unknown.

The structure of peptidoglycan displays some variability among the bacterial world, particularly in the peptide moieties. This variability is then observed in lipid II, which carries the peptidoglycan subunit [54]. Although the activity spectrum of ColM is restricted to *E. coli* strains or related species, the protein was able to hydrolyze lipids II with chemotypes characteristic of Gram-positive bacteria such as *Enterococcus faecalis*, *Enterococcus faecium* and *Staphylococcus aureus* (Table 2) [55]. These lipids contain L-lysine at position 3 instead of *meso*-A_2_pm found in most Gram-negative bacteria. Moreover, their L-lysine residue is substituted by peptides of different natures and lengths: L-Ala-L-Ala (*E. faecalis*), D-*iso*-Asn (*E. faecium*), Gly_5_ (*S. aureus*) [54].

### 2.4. Immunity

The production of the ColM immunity protein, Cmi, is performed concomitantly to that of ColM, in order to protect the ColM-producing cell against the effects of the toxin. Cmi (also called ImM), composed of 117 amino acids, is located in the periplasm while anchored to the plasma membrane by a transmembrane segment (residues 1–23) [56,57,58]. The presence of this anchor is not mandatory to confer immunity, since the expression of the soluble domain of Cmi within the periplasm was sufficient to protect the cells against the action of ColM [49,58]. However, the immunity was more efficacious when Cmi was anchored to the membrane.

Cmi exists in both monomeric and dimeric forms [59]. The structure of the soluble domain of Cmi in monomeric form has been solved in 2011 [49]. It is composed of four α-helices and four β-strands forming a β-sheet, which winds round a long central α-helix. This structure was stabilized by the presence of a disulfide bridge between the two only cysteine residues, C31 and C107 (Figure 8).

The structure of the dimeric form of Cmi has been published afterwards [59]. It was characterized by a domain exchange between the two protomers and the formation of two intermolecular disulfide bonds. The surface of dimerization is composed of the long central α-helix of one protomer and the β-sheet of the other protomer. The intermolecular disulfide bridges are formed between C31 of the first protomer and C107 of the second protomer (Figure 8). The formation of these bridges is essential for Cmi activity [49], since the individual or simultaneous mutation of the cysteinyl residues lead to a total loss of activity of the immunity protein. The formation of the disulfide bridges is dependent on the presence of DsbA, an oxidase localized in the periplasm. Cmi displays high amino acid sequence similarity with proteins of the YebF family, in particular at the level of the cysteines involved in the formation of the disulfide bridges [49].

Structural and mutagenesis studies allowed the identification, besides the two cysteines, of other essential residues for Cmi activity. Mutations E113A and Y114 A in the C-terminal part of Cmi abolished its immunity activity [49]. The simultaneous mutation of the negatively charged residues E81, D82 and E8 was also responsible for a loss of immunity [59]. The structure of ColM has revealed a cluster of positively charged residues at its surface [24]. Therefore, it has been suggested that ColM and Cmi may interact through electrostatic interactions through these respective clusters. Some data suggest a direct interaction between ColM and Cmi. Indeed, Cmi protects ColM from degradation by trypsin *in vivo*. It has also been shown that, in a strain in which the import system of ColM was overexpressed, the protection by Cmi decreased unless the immunity protein was overexpressed, thereby suggesting a titration of ColM by Cmi [57]. However, no genuine interaction was clearly demonstrated yet, and the exact mechanism of action of Cmi is still unknown.

Another type of immunity, mediated by the CbrA protein, has been identified in cells that produce neither Cmi nor ColM [60]. CbrA displays sequence similarities with geranylgeranyl diphosphate reductases and was hypothesized to function as a reductase of isoprenoid molecules [60]. The resistance to ColM of CbrA-overproducing *E. coli* cells was indeed shown to rely on a structural modification of the carrier lipid C_55_-P and its derivatives, including lipids I and II, namely the saturation of their α-isoprene unit. This unit is the closest one with respect to the ColM cleavage site, and it was demonstrated that this hydrogenation prevents the hydrolysis of the modified lipids by ColM, thereby explaining the acquisition of resistance [61].

## 3. The Orthologs of ColM

As mentioned above, several orthologs of ColM, which are produced by *Pseudomonas* spp., *Pectobacterium carotovorum*, *Burkholderia* spp. and *Klebsiella* species, have been identified [50,51,52,53,62,63,64,65,66,67,68,69]. These orthologs present a high sequence similarity with the C-terminal domain of ColM (Figure 9) and some of them have been biochemically and/or structurally characterized, as it will be discussed below.

A recent phylogenetic analysis of the carboxy-terminal domain sequences of ColM and some of its characterized and putative homologs has revealed that most of these bacteriocins can be affiliated to two main clades referred to as ColMα and ColMβ [53]. According to this study, most of the ColM family members, in particular those from enterobacteriaceae, pectobacteria and pseudomonads, belong to the ColMα clade. In spite of certain divergences, they possess a well-recognized ColM signature in their catalytic domain. On the contrary, the ColMβ clade comprises fewer members, notably those originating from *Burkholderia* species. This study also pointed out the diversity of the cognate immunity proteins expressed by the ColM-like-producing bacteria (see below the case of pseudomonads) [53].

### 3.1. From Pseudomonas *spp.*

#### 3.1.1. General Features

Different ColM orthologs from genus *Pseudomonas* spp. have been identified and characterized [50,62,67,69]. These are PaeM1, PaeM2, PflM and PsyM, that have been found in *P. aeruginosa* JJ692, *P. aeruginosa* NCTC10332, *P. syringae* pv. tomato DC3000 and *P. fluorescens* Q8r1–96, respectively. In some other *P. aeruginosa* strains, a distantly related *colM*-like gene, encoding an ortholog named PaeM4, has been also identified [68].

PaeM1, PaeM2, PflM and PsyM were shown to display the same activity of degradation of lipid II in vitro [50,62,69]. The PaeM variants display the highest specific activity among the different homologs tested, being 550-fold higher as compared to that of ColM. PsyM displays a specific activity of the same order of magnitude as that of ColM, while PflM is ca. 20-fold less active than ColM (Table 3). The enzymatic activity of the PaeM4 ortholog has never been tested.

Their respective activity spectra against *Pseudomonas* spp. are very restricted, as mentioned by several studies [50,51,65], and cross-cytotoxicity rarely occurs [50]. An illustration of this feature was recently made with the PaeM1 (from *P. aeruginosa* JJ692) and PaeM2 (from *P. aeruginosa* NCTC10332) proteins. Both variants, which share 90% of amino acid sequence identity, were tested on a high number of *P. aeruginosa* strains, including clinical isolates. Among the 65 strains tested, only 2 were susceptible to PaeM1, 5 to PaeM2 and 11 to both variants. The discrimination between both variants was correlated in particular to the sequence variations of their N-terminal domain, but also to the polymorphism of the outer membrane protein they target, the FiuA ferrichrome receptor (see below) [66,69].

None of these four ColM-like proteins from *Pseudomonas* display cytotoxicity against *E. coli*. However, they are capable of exerting a cytotoxic activity against *E. coli* when they are addressed into the periplasm through osmotic shock [62] or via the Sec system [71]. This shows that these proteins are potentially capable of eliminating other bacterial species than those naturally targeted as long as they reach their lipid II target. Moreover, PaeM is also able to hydrolyze in vitro lipids II of different chemotypes found in Gram-positive species [62], as it was shown for ColM (see above). It has also been demonstrated that PaeM displays a bacteriostatic effect when added to a *P. aeruginosa* DET08 culture [50], whereas a bacteriolytic effect is observed when ColM is added to a sensitive *E. coli* culture. 

#### 3.1.2. Comparison of 3D Structures

The structure of PsyM (also referred as Syringacin M) has been solved in 2012 [63], allowing to delineate its different domains (Figure 10). The translocation domain (residues 1–39) was totally disordered, as is the N-terminal domain of ColM. The involvement of this domain in the translocation of PsyM has been confirmed since the truncated forms PsyM Δ1–10 or PsyM Δ1–20 did not display any activity against *P. syringae*. In the same time, both variants inhibited the full-length PsyM in a competitive manner, thereby demonstrating that they still can bind the receptor. The binding to the outer membrane receptor is operated by the central reception domain (residues 40–127). Finally, the activity domain (residues 128–276) displays a high structural similarity with the equivalent domain of ColM (RMSD of 2.6 Å) [63].

This strong structural similarity is related to the high sequence similarity. A structural similarity is also observed between the respective reception domains of ColM and PsyM although no sequence similarity appears. Therefore, the structural superimposition of these two proteins on the global reception and activity domains leads to moderate structural variations with an RMSD of 3.3 Å. The resolution of the PsyM structure confirmed the necessity of a divalent cation for the catalytic process, since a Ca^2+^ ion was co-ordinated by D232 residue (corresponding to D226 in ColM). PsyM displayed the same enzymatic activity in the presence of Ca^2+^, Mg^2+^ or Mn^2+^, while the presence of EDTA in the reaction mixture inhibited the activity. The D232A mutation resulted in the loss of enzymatic activity and the structure of this variant showed the absence of Ca^2+^ ion in the active site. These data show the importance of residue D232 in the co-ordination of a divalent cation potentially involved in the binding of the pyrophosphate group of lipid II. Therefore, the ColM and PsyM proteins display strong functional and structural similarities, suggesting a common ancestor, which has evolved in a divergent manner in both genera, to target specific receptors [63].

The PaeM1 protein (also referred as Pyocin M1) has also been the subject of a structural and functional study [62] which assigned residues 1 to 32 to the N-terminal translocation domain, residues 33–136 to the central domain, and residues 137 to 289 to the C-terminal domain. The latter domain is very similar to that of ColM, exhibiting a 2.9 Å RMSD over the superposition of 138 Cα atoms. As it is the case for PsyM, this structural similarity is related to the sequence similarity between the catalytic domains of ColM and PaeM1.

In spite of this strong similarity, important structural differences exist between PaeM1 and ColM, particularly in their respective catalytic sites. The lateral chain of residue D241 in PaeM1, which co-ordinates a magnesium ion, points towards the inner part of the active site whereas the lateral chain of D226 in ColM protrudes towards the outside of the protein. Therefore, these two lateral chains have totally opposed orientations. Moreover, the lateral chain of D241 is superimposed with the lateral chain of Y228 in ColM. These differences in the orientations of the lateral chains of the conserved residues (D241, Y243, D244, H250, R251) is probably one of the main factors explaining the difference in the in vitro activities of ColM and PaeM1: the conformation of the active site of PaeM1 would be closer to the optimal conformation of the active site. However, the lateral chains of certain conserved residues are not oriented towards the core of the active site: this suggests that the observed conformation does not represent the final conformation of the active site. The active site probably undergoes further conformational rearrangements following the binding of the substrate. Moreover, the isolated domains of PaeM1 and ColM display a much higher activity of degradation of lipid II in vitro as compared to the respective full-length proteins [50,62]. This suggests that the presence of the reception domain, which is in interaction with the catalytic domain, maintains the protein in an only partially active form.

#### 3.1.3. Diversity of Immunity Proteins

Studies based on in silico analysis of different genomes of *Pseudomonas* strains producing bacteriocins orthologues to ColM, have identified genes encoding three different types of immunity proteins [53,65]. Depending on the species, immunity seems to be provided either by an inner membrane-anchored periplasmic protein or an integral membrane protein. The encoding genes are located downstream of the gene encoding the bacteriocin. 

The first type of immunity gene codes for proteins of ca. 137 residues in average, called PmiA (*Pseudomonas* colicin M
immunity type A). These immunity proteins are unrelated in sequence and topology to the *E. coli* Cmi protein. They are constituted of a predicted cleavable signal sequence either of Tat or Sec type, followed by three transmembrane segments. When expressed in sensitive strains, the genes corresponding to the immunity proteins of PaeM, PflM and PsyM conferred immunity towards their cognate bacteriocins, thus confirming their role [65,69]. Moreover, despite the absence of conserved amino acid sequences, these immunity proteins displayed cross-immunity protection on various *Pseudomonas* strains [65].

The less widespread second type of immunity protein has been isolated after the analysis of 14 *Pseudomonas* strains, mainly *P. syringae* strains. The identified gene encodes a protein containing ca. 100 residues called PmiB (*Pseudomonas* colicin M
immunity type B). Two conserved cysteine residues have been found among the different PmiB sequences, and this is quite reminiscent of Cmi [53]. These proteins possess a putative N-terminal cleavable signal sequence, which may lead to the release of the proteins into the periplasm after the translocation through the inner membrane. It is noteworthy that PmiB-encoding genes are absent from *P. aeruginosa* genomes.

Finally, a third type of immunity protein, called PmiC (*Pseudomonas* colicin M
immunity type C) has been identified and predicted to be constituted of six transmembrane α-helices. The PmiC-encoding genes apparently occur only in *P. aeruginosa* genomes and the corresponding proteins were shown to confer immunity against PaeM4 [72].

#### 3.1.4. Hybrid Proteins Exhibiting ColM Catalytic Domain

*In silico* analysis of *Pseudomonas* spp. genomes has allowed the identification of new proteins possessing a domain similar to the catalytic domain of ColM [66]. Besides, they possess a second domain similar to the ionophoric domain of colicin N. These proteins, identified in several *Pseudomonas* species, in particular *P. fluorescens*, *P. libanensis*, *P. synxantha*, have been called PmnH (*Pseudomonas* colicin M colicin N
Hybrid). Their coding sequences share an almost perfect sequence identity (98%) with the exception of the PmnH protein from *Pseudomonas* spp. 25R14 (58%). This type of bacteriocins possessing two catalytic domains had already been identified in *Pseudomonas*, but exhibiting two catalytic domains of similar function, such as the PsdH1 and PsdH2 proteins that have two DNase domains [66]. PmnH possesses an N-terminal short proline-rich sequence similar to that already identified in PsyM [51] and BurM (a homologue of ColM produced in *Burkholderia*, see below) [52]. This sequence is probably involved in the translocation of the protein across the outer membrane through its interaction with a TonB homolog. The cytotoxicity of PmnH has been tested against 35 *Pseudomonas* strains, out of which 11% came out to be susceptible. Their susceptibility was increased in an iron-deprived medium. This characteristic has allowed the identification of the FiuA protein, a TonB-dependent transporter of ferrichrome, as the receptor of PmnH [73]. FiuA has also been shown to be parasitized by PaeM1, PaeM2 and PflM [66]. 

The cytotoxic activity of PmnH is apparently due to the ionophoric domain only, since the cytotoxic activity was no longer observed upon deletion of the latter domain. The domain homologue to ColM thus seems to be devoid of activity. The identification of ColM-like bacteriocins has allowed to define a conserved motif in the active site: DxYD(x5)HR. This motif somewhat varies in PmnH: HxYD(x5)FK. The D226 residue of ColM is replaced by H224 in PmnH. It has been shown that any mutation of D226 in ColM abolishes the activity of the protein [26,28]. Similarly, the essential catalytic R236 residue of ColM is replaced by K234 in PmnH, which could explain the lack of activity of this domain.

An immunity gene associated to *pmnH* has been identified and the capability of the corresponding protein to protect against PmnH has been demonstrated. The topology of this protein, ImnH, was predicted to contain a signal sequence of Sec type followed by three transmembrane segments, similar to PmiA topology. However, PmiA and ImnH display only a low sequence homology. The immunity protein of colicin N is also composed of four transmembrane segments and shares a 20% sequence identity with ImnH, suggesting a protective effect of ImnH against the activity of the ionophoric domain of PmnH [66].

### 3.2. From Burkholderia *spp.*

Burkhocins M (BurM), orthologs of ColM produced by *Burkholderia* spp., have been identified among 16 *Burkholderia* strains through in silico genomic analyses [52]. They are characterized by the presence of an N-terminal signal sequence involved in their secretion. In the N-terminal part of the proteins, a conserved sequence corresponding to a TonB box has been found; this sequence is specific to *Burkholderia*.

Immunity proteins displaying a topology similar to PmiA or PmiB have been identified in some *Burkholderia* spp. Strains [52]. Thus, the BmiA protein (Burkhocin M
immunity type A) displays a membrane topology similar to PmiA, exhibiting three transmembrane segments but no signal sequence. The topology of BmiB (Burkhocin M
immunity type B) is similar to that of PmiB with an N-terminal membrane anchor. The genes encoding these immunity proteins can be located downstream or upstream of the BurM-encoding genes. 

The BurM1 and BurM2 proteins, produced by *B. ambifaria* MEX-5 and *B. ambifaria* AMMD, respectively, have been tested against 44 *Burkholderia* strains, out of which 26% were susceptible to BurM1 and 7% to BurM2. Immunity tests revealed no cross-immunity, as the BmiA-type immunity protein associated to BurM1 only protects against the action of BurM1, whereas the BmiB-type immunity protein associated to BurM2 only protects against the action of BurM2 [52].

All burkhocins identified to date possess a ColM-type catalytic domain carrying the characteristic DxYD(x5)HR motif sequence required for the hydrolase function. However, none of them has been tested yet for lipid II-degrading activity.

### 3.3. From Pectobacterium carotovorum

Two ColM orthologs have been identified in *Pectobacterium carotovorum* [51]: pectocins M1 (PcaM1) and M2 (PcaM2) from *P. carotovorum subspp. carotovorum PC1* and *P. carotovorum subspp. brasiliensis* BPR1692, respectively. Their translocation and reception domains are formed of a ferredoxin domain containing a [2Fe-2S] cluster. This domain displays 60% of sequence identity with spinach ferredoxin. Their catalytic domain itself displays 46% of sequence identity with the catalytic domain of ColM. The ferredoxin and catalytic domains are connected through an α-helix linker. PcaM1 and PcaM2 proteins, which share 58% sequence identity, have a restricted spectrum of action, targeting *P. carotovorum* and *P. astrosepticum* strains [51]. Both proteins possess a lipid II-degrading activity [51,74]. PcaM1, as well as its isolated catalytic domain, were also demonstrated to inhibit *E. coli* cell growth when directly addressed into the periplasm via the Sec system [74].

A gene potentially encoding an immunity protein (the corresponding product presenting 24% of sequence identity with Cmi) has been identified in *P. carotovorum subspp. carotovorum PC1* genome directly downstream of the PcaM1-encoding gene, whereas no immunity protein associated to PcaM2 has been identified yet [51]. 

Since both Pectocin M proteins possess a ferredoxin domain, their cytotoxic activity has been tested in iron-deprived media [51]. In these conditions, the cytotoxic activity is increased with respect to the activity observed in a non-deprived medium. Moreover, the cytotoxic activity of these bacteriocins is inhibited in the presence of a high concentration of spinach ferredoxin in the medium, thereby demonstrating that PcaM1 and PcaM2 utilize a receptor involved in the import of iron [51]. It has been shown that both proteins parasitize the same receptor, identified as the FusA protein, a TonB-dependent protein [75]. The structure of this receptor has been solved, and as all the TonB-dependent receptors from *E. coli*, it forms a 22 strand-containing β-barrel. The pore of the barrel is filled by an N-terminal plug. The extracellular loops of FusA present a unique organization mainly structured in β-sheet. The structure of a complex formed by purified FusA and isolated ferredoxin domain of PcaM1 has been solved by NMR and has suggested a 1:1 stoichiometry [75].

The structure of PcaM2 has been solved (Figure 10) [64]. Contrary to ColM and its other known orthologs, PcaM2 does not present an intrinsically disordered N-terminal domain. This suggests that PcaM1 and PcaM2 use a translocation mechanism that differs from ColM and other orthologs. The presence of the α-helix linker between the ferredoxin and catalytic domains offers some flexibility to PcaM2. Small angle X-ray scattering experiments have shown that this protein exists as two conformations in solution, a bended form and a stretched form. The structure of the ferredoxin domain of PcaM2 is almost identical to that of spinach ferredoxin (RMSD of 0.90 Å). The catalytic domain of PcaM2 is structurally very close to the catalytic domain of ColM (RMSD, 1.7 Å). The organization of the active site is similar to that found in ColM, except for residue R236 whose lateral chain is oriented towards the inside of the active site, contrary the lateral chain of residue R236 of ColM, which rises up towards the outside of the protein [64].

The structures of FusA and PcaM2 have been used to model the docking of the pectocin to its receptor [75]. It appears that the flexibility of PcaM2, conferred by its α-helix linker, is essential for the first steps of the binding. These docking experiments also suggest that a movement of the extracellular loops occurs during the binding of PcaM2 so that these intracellular loops may surround the ferredoxin domain of the pectocin. During the binding to the receptor, the passage from the bended form to the stretched form would allow the extracellular loops to surround the ferredoxin domain.

### 3.4. From Klebsiella *spp.*

Genes encoding ColM-like bacteriocins (klebicins) have also been identified in silico in *Klebsiella* genomes [53]. BLAST search followed by amino acid sequence alignment led to the identification of four putative ColM-like proteins: KpneM from *K. pneumoniae* EWD35590.1, KpneM2 from *Klebsiella* sp. WP_047066220, KvarM from *K. variicola* CTQ17225.1 and KaerM from *K. aerogenes* WP_015367360.1, displaying 48, 42, 43 and 29% sequence identity with ColM, respectively [67].

Klebicin-encoding genes have been overexpressed in *Nicotiana benthamiana* plants. Crude plant extracts exhibit broad antimicrobial activity against different *Klebsiella* species. This shows that, in contrast to other bacteriocins (e.g., pyocins) which are species-specific, klebicin activity is rather genus-specific.

The three most active klebicins (KpneM, KpneM2 and KvarM) have been selected for purification and further experiments. MIC values for *Klebsiella* susceptible strains are in the range 0.1–0.8 µg/mL. Broad-spectrum activity was found for KvarM and KpneM against a panel of 100 *K. pneumoniae* and *K. oxytoca* strains resistant to at least one antibiotic, some of them being multi-drug resistant (but all carbapenem sensitive); similarly, they were active against a more restrictive panel (7) of carbapenem-resistant *K. pneumoniae* strains. In both cases, KpneM2 displayed a narrower spectrum of activity.

KvarM exhibits high antimicrobial activity on planktonic and biofilm cells of *Klebsiella* species, reducing the CFU number by 2–3 logs in both cases.

Transposon mutagenesis experiments showed that KpneM, KpneM2 and KvarM require the ferrichrome transporter FhuA for reception, and TonB and ExbB for translocation. Although no in vitro experiments on lipid II were performed, sequence homology strongly suggests that this lipid is the target of these klebicins [67].

## 4. Possible Applications and Uses

Facing the emergency of the drug resistance situation worldwide, the panel of available and suitable antimicrobial molecules has to be enlarged. In this respect, bacteriocins are considered as potential antimicrobial therapeutic agents. 

However, to date, the only application field of bacteriocins is agrobusiness, but not in an extensive way, as nisin, produced by the Gram-positive *Lactococcus lactis* 6F3 strain, is the only one substantially used as food preservative [76]. Nevertheless, using plant-based expression systems, several recombinant colicins have been described to control the foodborne multidrug-resistant *E. coli* O104:H4 serotype [77,78,79]. Moreover, among the many colicins tested in these studies, ColM was demonstrated to be the most broadly active one for the control of the seven major foodborne enterohemorrhagic *E. coli* (EHEC) strains, including the O157:H7 serotype. In addition, the use of selective cocktails of two colicins (for example M + E7, [77]) or more (for example M + E7 + Ia + 5 + K + U, [79]) with complementary or synergistic activities showed significant decreases of the bacterial populations, in food products, of certain EHEC and Shiga-toxin producing *E. coli* strains, respectively. These promising investigations led the NOMAD BIOSCIENCE Ltd. to apply for the GRAS status for its COLICIN cocktail, constituted of ColE1, ColE7, ColIa, ColN, ColK, ColU, Col5, ColB and ColM. This colicin mixture is expected to protect fruits and meat products from food-borne pathogens [80].

Another area where bacteriocins may offer a potential alternative to traditional antibiotics is obviously human health. The human body is constantly and naturally exposed to such toxins, either through food product intake (lactic acid bacteria from dairy products producing a large panel of bacteriocins) or from its commensal microflora (as a large proportion of *E. coli* strains produce colicins) [81]. Due to their narrow antibacterial spectra, ColM-like proteins represent reliable candidates, as each of them will target specific bacterial strains. In this respect, if we consider using them as therapeutic agents, the way they could be inoculated will depend on the cell tissue to be treated. Accordingly, the ColM orthologs from *P. aeruginosa*, PaeM1 and PaeM2, could be delivered by nebulization in cystic fibrosis suffering patients. On the other hand, the ColM protein could be used to cure gastrointestinal infections and be taken by oral route, assuming that it will not be entirely broken down by digestive enzymes. Moreover, in order to enlarge the antibacterial spectra, chimera-ColM like proteins produced by engineering experiments could be tested. Accordingly, Latino and collaborators recently showed that the differences in the antibacterial spectra of both PaeM variants on 65 *Pseudomonas* clinical strains were mainly due to sequence variation in the N-terminal moiety of both proteins. The design of chimeras, by partial or total replacement of the N-terminal translocation region of one homolog by the corresponding region of the other, led to the generation of chimeric proteins possessing antibacterial activity against *Pseudomonas* strains previously resistant to one or another natural PaeM homolog [69].

A last point that should be taken into consideration, concerning the use of ColM-like proteins, is their likely absence of deleterious effects in human or animal cells. Bacteriocins generally display low toxicity towards eukaryotic cells [82]. ColM-like proteins target polyprenyl phosphate lipids that also exist in eukaryotes, as these lipids allow the translocation of sugar molecules across cell membranes in all kingdoms of life. However, the structure of the lipid carrier varies to some extent in living organisms, regarding the length of the carbon chain, its stereochemistry and the hydrogenation status of the α-isoprenyl unit. This last difference is of great importance as it was recently demonstrated that the ColM protein was totally unable to cleave dihydro-lipid II and presumably dolichyl-phosphate-linked glycans found in human [61], opening the way to a the potential utilization of these colicins as new antibacterial agents.

## 5. Concluding Remarks

Among bacteriocins interfering with peptidoglycan metabolism, ColM and its orthologs are particular as their mode of action consisting in lipid II cleavage is unique among colicins. Targeting such an essential bacterial component, present in all types of bacteria, potentially makes the ColM-like proteins “universal” antibacterial agents. This feature clearly opens the way towards an exploitation of these proteins as broad-spectrum antibacterial agents, depending “only” on their capabilities to reach their target, protected either by an outer membrane in Gram-negative bacteria or by a thick peptidoglycan layer in Gram-positive ones. Therefore, two current challenges now prevail: (i) the understanding of the underlying interactions occurring between the ColM-like proteins and their molecular partners required for their cytotoxicity, and (ii) the design of chimeric ColM-like proteins by molecular engineering in order to adapt the antibacterial spectra of these proteins and increase the availability of efficient molecules to fight antibiotic-resistant pathogens.

## Figures and Tables

**Figure 1 antibiotics-10-01109-f001:**
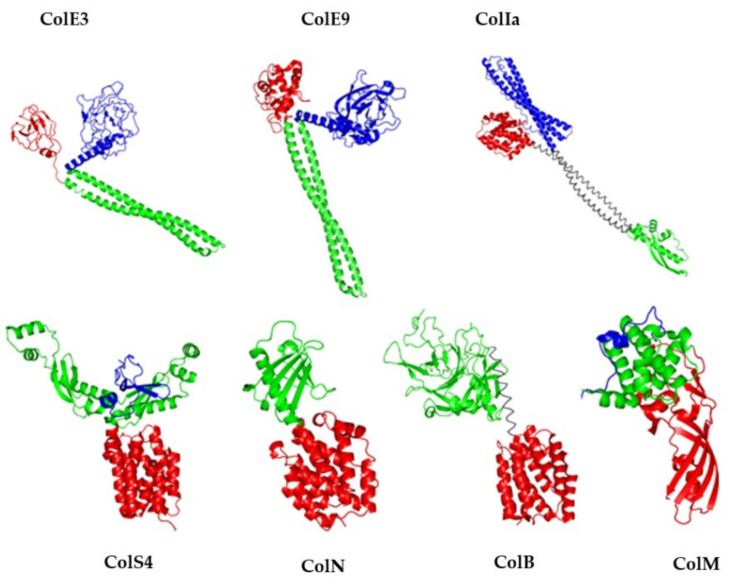
Modular organization of colicins. 3D-structures of ColE3 (PDB entry: 1JCH), ColE9 (PDB entry: 5EW5); ColIa (PDB entry: 1CII); ColS4 (PDB entry: 3FEW); ColN (PDB entry: 1A87); ColB (PDB entry: 1RH1); ColM (PDB entry: 2XMX). For all structures, the translocation domain is colored in blue, the reception domain in green and the activity domain in red. The figure was prepared with the atomic coordinates from the PDB by using PyMOL (DeLano Scientific).

**Figure 2 antibiotics-10-01109-f002:**
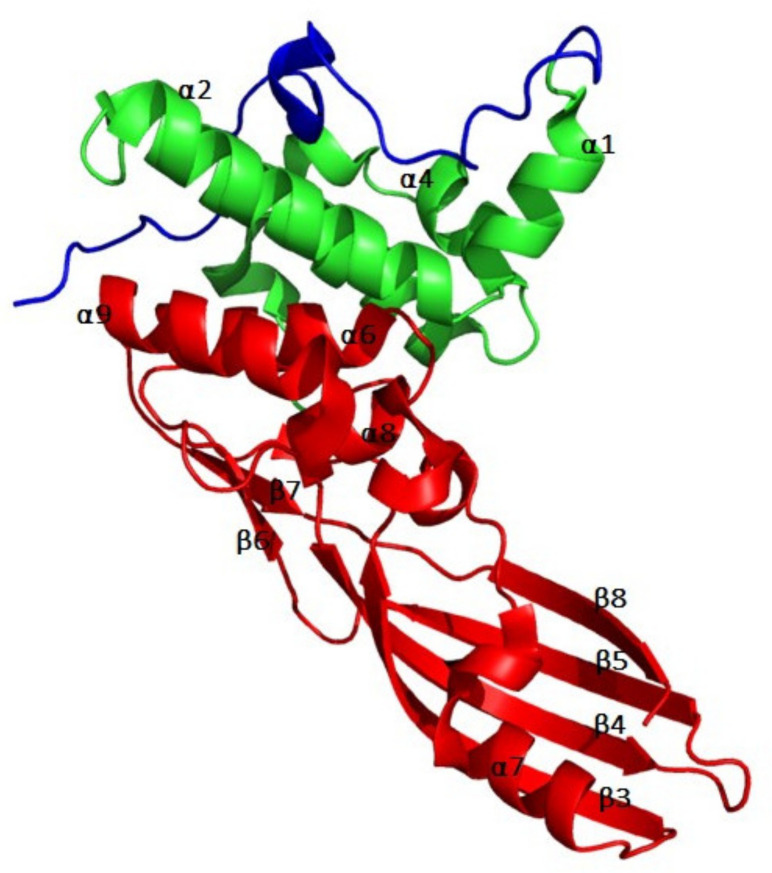
3D structure of colicin M. The three domains of the protein are colored differently, according to [26]: blue for the translocation domain, green for the reception domain, and red for the catalytic domain. Secondary structure elements are numbered according to their order of appearance within the primary structure of the protein. The figure was prepared with the atomic coordinates from the PDB (PDB entry: 2XMX) by using PyMOL (DeLano Scientific).

**Figure 3 antibiotics-10-01109-f003:**
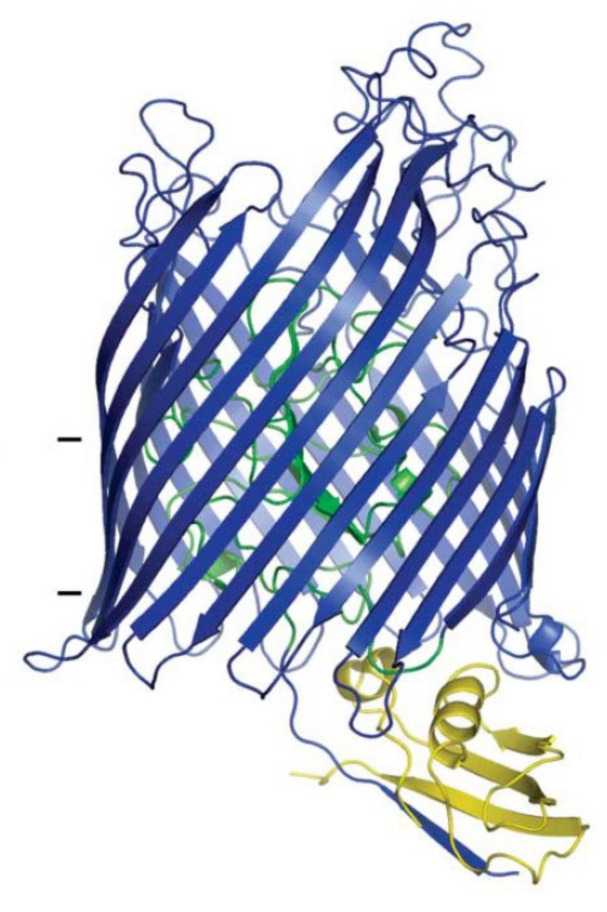
Structure of the complex formed between FhuA and TonB. The proteins FhuA and TonB are represented in cartoon. The C-terminal domain of TonB (residues 158–235) is colored in yellow. The plug domain of FhuA (residues 19–160) is colored in green. The rest of the molecule (residues 8–18 and 161–725) is colored in blue. Horizontal bars delineate approximate outer membrane boundaries. [Reproduced with permission from Pawelek et al. (2006) [34] and agreement from the American Association for the Advancement of Science and Copyright Clearance Center].

**Figure 4 antibiotics-10-01109-f004:**
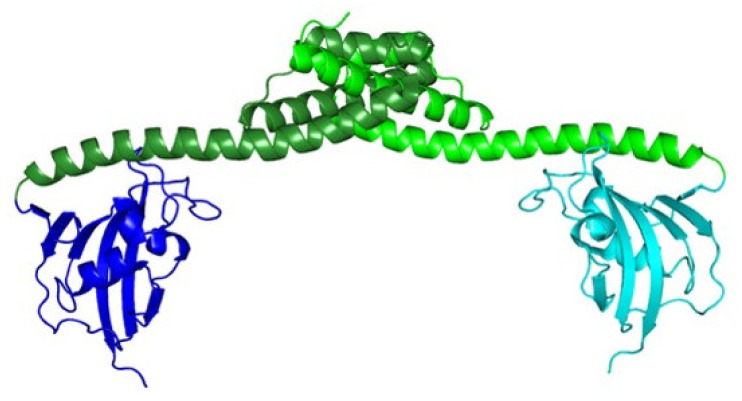
3D structure of FkpA as a dimer. The structure of each protomer is represented in cartoon. The chaperone domains of both protomers are represented in light green and dark green, respectively. Their PPIase domains are represented in turquoise and dark blue, respectively. The dimerization interface is localized at the level of the chaperone domains (PDB entry: 1Q6U). The figure was prepared with the atomic coordinates from the PDB by using PyMOL (DeLano Scientific).

**Figure 5 antibiotics-10-01109-f005:**
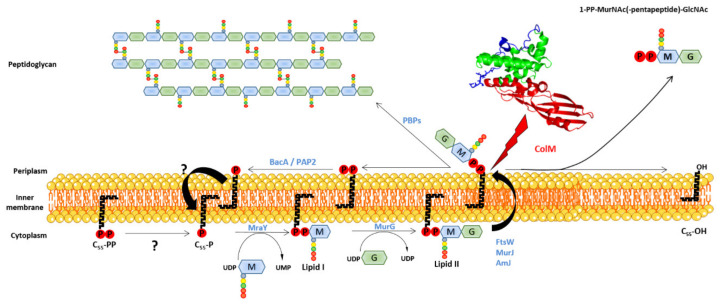
Peptidoglycan biosynthesis pathway and mode of action of ColM. ColM hydrolyzes the phosphodiester bond of lipid II (red lightning), thereby leading to the formation of two products: undecaprenol (C_55_-OH) and 1-PP-MurNAc(-pentapeptide)-GlcNAc. ColM is represented in cartoon.

**Figure 6 antibiotics-10-01109-f006:**
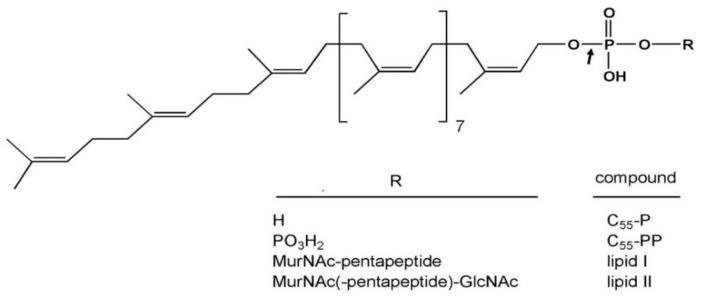
Structure of the C_55_-P carrier lipid derivatives. The arrow indicates the site of cleavage by ColM in the peptidoglycan lipid I and lipid II intermediates.

**Figure 7 antibiotics-10-01109-f007:**
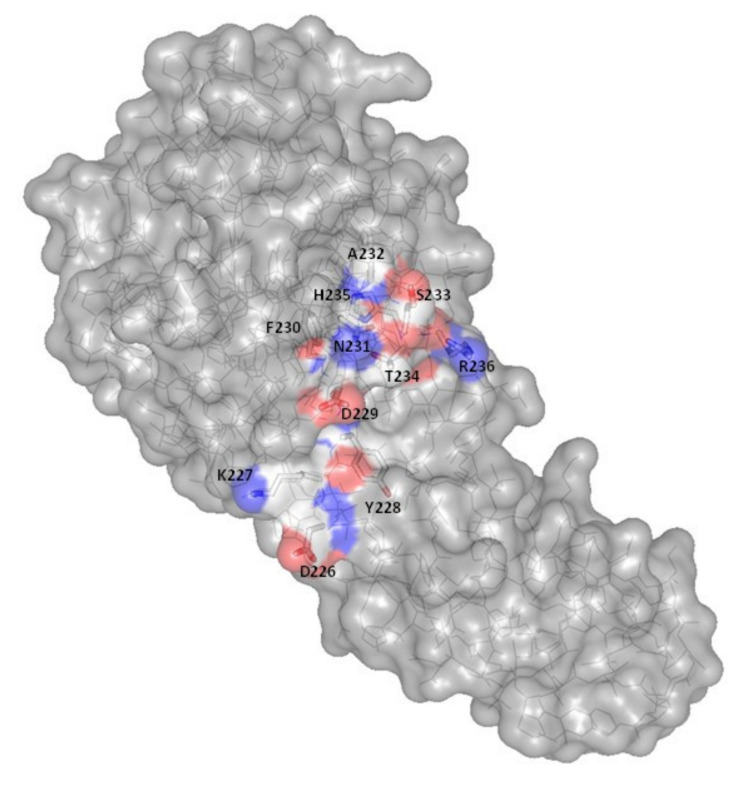
Localization of the active site residues on the surface of ColM. Representation in van der Waals surface of the D^226^KYDFNASTHR^236^ residues, localized at the surface of ColM. For these residues, a color code has been attributed to the atoms: carbon in white, nitrogen in blue and oxygen in red. The residues are identified through their position in the primary structure as well as through the corresponding one-letter code. The figure was prepared with the atomic coordinates from the PDB (PDB entry: 2XMX) by using PyMOL (DeLano Scientific).

**Figure 8 antibiotics-10-01109-f008:**
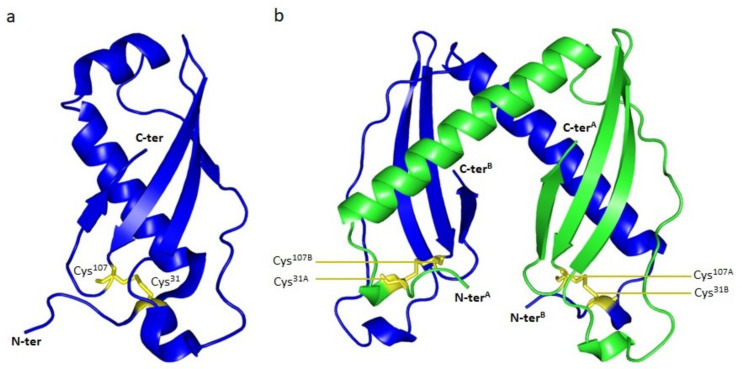
3D-structures of Cmi. (**a**) Structure of Cmi as a monomer (PDB entry, 2XGL). The N-terminal (N-ter) and C-terminal (C-ter) ends are indicated. The cysteinyl residues 31 and 107 (Cys31 and Cys107), responsible for the formation of the disulfide bond, are colored in yellow and represented as sticks. (**b**) Structure of Cmi as a dimer (PDB entry, 4AEQ). Each protomer is colored differently (green for protomer A and blue for protomer B). The N-terminal (N-ter^A^ and N-ter^B^) and C-terminal (C-ter^A^ and C-ter^B^) of both protomers are indicated. The residues involved in the formation of intermolecular disulfide bridges are identified. Residues Cys^31A^ and Cys^107B^, belonging to monomers A and B, respectively, form the first disulfide bridge. Residues Cys^31B^ and Cys^107A^ form the second one. The structures of Cmi as monomer and dimer are represented in cartoon. The figure was prepared with the atomic coordinates from the PDB by using PyMOL (DeLano Scientific).

**Figure 9 antibiotics-10-01109-f009:**
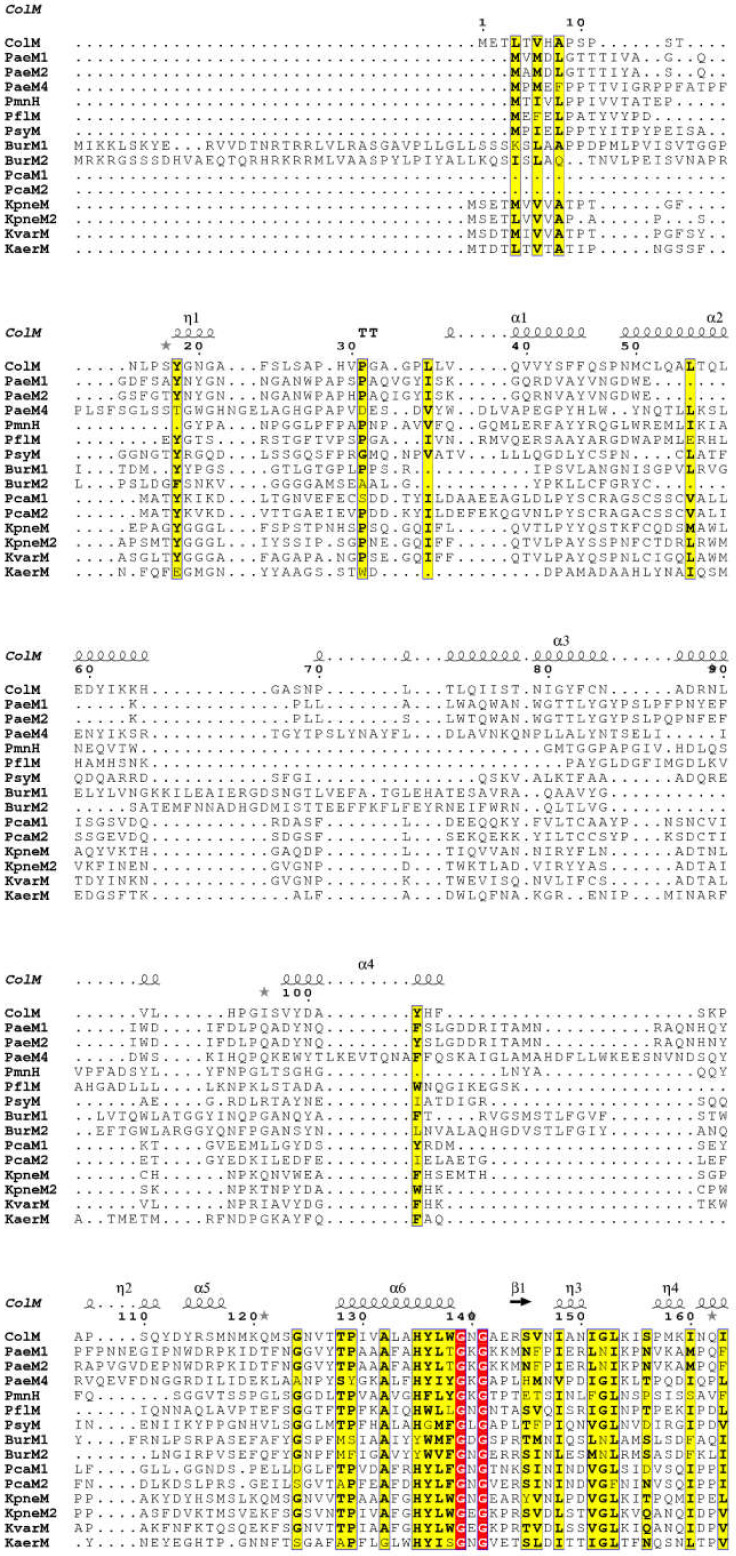
Sequence alignment of ColM and its homologs. The strictly conserved residues are highlighted in red, and the conserved residues in yellow. The strong sequence homology identified in the C-terminal part is boxed in red. The secondary structures of ColM are represented above the alignment. The amino acid residues essential for ColM activity are indicated in red below the sequences. This sequence alignment was generated with the ClustalW program and visualized thanks to the ESPript program [70].

**Figure 10 antibiotics-10-01109-f010:**
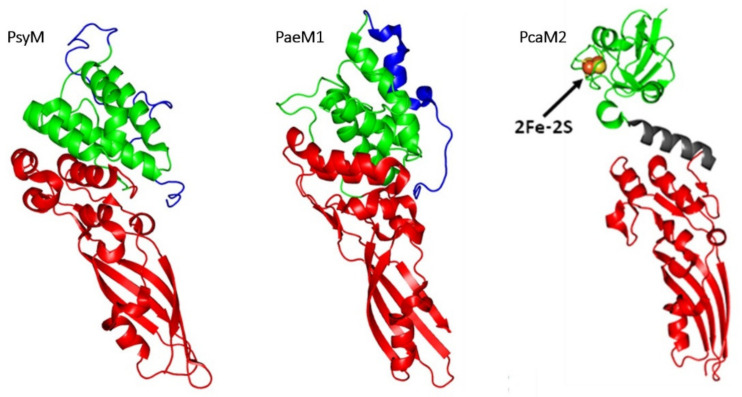
3D-structures of ColM-like proteins. PsyM (PDB entry: 4FZM), PaeM1 (PDB entry: 4G76), PcaM2 (stretched form; PDB entry: 4N58). For all structures, the translocation domain is colored in blue, the reception domain in green and the activity domain in red. 2Fe-2S: [2Fe-2S] cluster. The figure was prepared with the atomic coordinates from the PDB by using PyMOL (DeLano Scientific).

**Table 1 antibiotics-10-01109-t001:** In vitro and in vivo activity of different mutants of ColM. ND, not detected up to 10 µg of purified protein (after [26]).

Protein	Enzymatic Activity (pmol·min^−1^·mg^−1^)	Cytotoxicity ^a^ (ng)
Wild-type	630 ± 57 (100%)	0.4
D226A	2 ± 0.9 (0.3%)	>10,000
Y228A	ND	>10,000
D229A	12 ± 5 (1.9%)	>10,000
H235A	9 ± 4 (1.4%)	>10,000
R236A	ND	>10,000

^a^ The cytotoxicity has been tested by spotting a certain amount of protein (in ng) on a cell overlay of *E. coli* BW25113.

**Table 2 antibiotics-10-01109-t002:** Activity of ColM on ramified lipids II. The peptidic nature of the ramification is indicated in bold. The bacterial species corresponding to each lipid II is indicated in parentheses (after [55]).

Composition of the Peptide Part of Lipid II	SA ColM (%) ^a^
L-Ala-γ-D-Glu-*meso*-A_2_pm-D-Ala-D-Ala (Gram-negative)	100 ^b^
L-Ala-γ-D-Glu-L-Lys-D-Ala-D-Ala (Gram-positive)	90 ± 6
L-Ala-γ-D-Glu-L-Lys-**(L-Ala)**-D-Ala-D-Ala (*E. faecalis*)	87 ± 11
L-Ala-γ-D-Glu-L-Lys-**(L-Ala)_2_**-D-Ala-D-Ala (*E. faecalis*)	77 ± 5
L-Ala-γ-D-Glu-L-Lys-**(D-iAsn)**-D-Ala-D-Ala (*E. faecium*)	94 ± 8
L-Ala-γ-D-Glu-L-Lys-**(Gly)**-D-Ala-D-Ala (*S. aureus*)	79 ± 8
L-Ala-γ-D-Glu-L-Lys-**(Gly)_3_**-D-Ala-D-Ala (*S. aureus*)	89 ± 8
L-Ala-γ-D-Glu-L-Lys-**(Gly)_5_**-D-Ala-D-Ala (*S. aureus*)	81 ± 11

^a^ The relative specific activity of ColM has been calculated from that measured for the *meso*-A_2_pm-containing lipid II, natural target of ColM, and expressed as the reference (100%). ^b^ Corresponds to 0.4 nmol·min^−1^·mg^−1^.

**Table 3 antibiotics-10-01109-t003:** Specific activities of ColM and its homologues PaeM1, PflM and PsyM originating from *P. aeruginosa*, *P. fluorescens* and *P. syringae*, respectively (after [50]).

Protein	Specific Activity (nmol·min^−1^·mg^−1^)
ColM	0.4
PaeM1	219
PflM	0.017
PsyM	1.1

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
