# Peer review of "The Biology of Colicin M and Its Orthologs"

_antibiotics, 2021, doi:10.3390/antibiotics10091109_

Round 1
Reviewer 1 Report
This is an excellent and comprehensive review, and provides and excellent short course on ColM and its orthologs. The figures are excellent, and provide significant depth to the writing in the text. The references are also very diverse in their publication date, where most are more than about 5 years old. There are some more recent references though, which makes this review highly relevant.
Overall the writing is excellent and clear, but there are some minor cosmetic errors throughout. Therefore the manuscript should go through another round of proofreading and editing before publication. For example, line 296 is missing a parenthesis, and lines 325 and 326 have the phrase "numerous evidences allow to conclude that". Line 770, "meet" should be "meat". There are many additional minor problems.
Reviewer 2 Report
This manuscript flows well and provides extensive information for the field. I have no detailed comments this time.
Reviewer 3 Report
Colicin M (ColM) is a toxin protein produced by E. coli. Compared with other colicin types, it has a unique antibacterial mechanism — interfering with the biosynthesis of peptidoglycan. This review systematically reviews the current knowledge of ColM and its orthologs including activity, mechanism of action, sequence structure and other known information and their potential use in food preservatives or antibacterial therapeutics. This review is well written and should attract interest from readers of Antibiotics. Therefore, I suggest publishing this manuscript with minor revisions.
"E. coil" in lines 11 and 14 of the summary on page 1 should become "E. coli"
The font format of “Similar experiments carried out with ColM did not lead 162 to” in line 162 on page 5 is inconsistent with other font formats.
Please increase the resolution of Fig 5 on page 8
The picture of Fig 8 on page 13 has a low resolution, which makes the position identification of the amino acid residues in the picture unclear.
Please unify the expressions of NAc(-pentapeptide)-GlcNAc and NAc-pentapeptide-GlcNAc on pages 8 and 9
Please check page 492, "PflM displays a specific activity of the same order of magnitude as that of ColM, while PflM is ca. 20-fold less active than ColM (Table 3)." is correct. It contradicts the content of Table 3.
Reviewer 4 Report
The manuscript reviewed the structures and functions of colicin M (ColM) and its orthologs comprehensively, such as narrow antibacterial spectra and enzymatic activity.
Comment 1
FhuA or other partners were confirmed to binding with ColM. Authors should provide the complex structure data to illustrate interactions.
Comment 2
In the manuscript, authors stated that ColM and its orthologs, such as PasM, PflM and PsyM etc., have a common ancestor. Then, authors should consider the evolutionary relationship about ColM and its orthologs.
Comment 3
As ColM immunity protein, Cmi and its orthologs should be considered about evolutionary relationship and interaction with ColM or its orthologs in this manuscript.
